# New Red-Shifted 4-Styrylcoumarin Derivatives as Potential Fluorescent Labels for Biomolecules

**DOI:** 10.3390/molecules27051461

**Published:** 2022-02-22

**Authors:** Raquel Eustáquio, João P. Prates Ramalho, Ana T. Caldeira, António Pereira

**Affiliations:** 1Hercules Laboratory, University of Évora, Largo Marquês de Marialva 8, 7000-809 Évora, Portugal; raqueleustaquio98@hotmail.com (R.E.); atc@uevora.pt (A.T.C.); 2Chemistry Department, School of Sciences and Technology, University of Évora, Rua Romão Ramalho 59, 7000-671 Évora, Portugal; jpcar@uevora.pt; 3Laqv-Requimte, University of Évora, Rua Romão Ramalho 59, 7000-671 Évora, Portugal

**Keywords:** coumarin, 4-styrylcoumarin derivatives, fluorescent labels, biomolecules labeling, large Stokes shifts

## Abstract

Important scientific areas, such as cellular biology, medicine, pharmacy, and environmental sciences, are dependent on very sensitive analytical techniques to track and detect biomolecules. In this work, we develop a simple, low-cost and effective synthetic strategy to produce new red-shifted 4-styrylcoumarin derivatives as promising inexpensive fluorescent labels for biomolecules. The extension of the delocalized π-electron system results in bathochromic shifts in these new coumarin derivatives, which also present large Stokes shifts. In addition, density functional theory and time-dependent density functional theory calculations helped to rationalize the photophysical properties observed by the experimental results.

## 1. Introduction

Cellular biology, medicine, pharmacy and environmental sciences require highly sensitive analytical techniques to track and detect nucleic acids, oligonucleotides, antibodies, amino acids, proteins, lipids, carbohydrates, and other biomolecules [1,2,3,4,5,6]. Most available techniques generally require labelling with a sensor, as fluorescent labels [7,8] electrochemical sensors [9], photochromic compounds [10], photo switchable biomaterials [11], colorimetric biosensors [12], radioactive tracers [13] and isotope markers [14]. Of all sensitive analytical techniques, fluorescent labelling, taking into account the high sensitivity of the fluorescence technique and its non-destructive nature, presents numerous advantages as it allows the use of small sample quantities as well as the respective fluorescent labels [15]. The availability and the development of new fluorophores are now enabling previously impossible studies of cellular processes and the detection of specific components of complex biomolecular assemblies with selectivity and exquisite sensitivity, in vitro and in vivo, as well the analysis of their interactions [16,17,18]. Fluorescent labels offer many advantages as they are highly sensitive, even at very low concentrations, and can form covalent linkages with the sample to be analyzed, producing stable bioconjugates [19]. They should be chemically stable and small in size, with insignificant interference on the biological functions and structure of the unlabeled biomolecules, producing high fluorescence quantum yield bioconjugates. The amine-reactive fluorescent labels, since amino groups are either abundant or easily introduced into biomolecules, are the most frequently used to prepare numerous bioconjugates for direct or indirect immunochemistry, fluorescence in situ hybridization (FISH), histochemistry, cell tracing, receptor binding and other biological applications [20,21,22]. In this context, due to the high cost of the available commercial fluorescent labels, coumarin derivatives can be a solution to develop low-cost new fluorophores with absorption and emission at long wavelengths, combined with large Stokes shifts. Coumarins, beyond their very significant pharmacological activity [23,24], are interesting alternatives for applications in research focused on emission layers in organic light-emitting diodes [25], solar cells dyes [26], nonlinear optical chromophores [27], fluorescent brighteners [28], fluorescent whiteners [29], fluorescent labels for physiological measurement [30], and more recently in labelling [31] and in caging [32]. The developments of the last decades evidenced that the substitution pattern and the nature of the substituents in the coumarin rings involving donor-π bridge-acceptor structures (D-π-A) promote the delocalization of the conjugated π electron system, producing derivatives with extraordinary spectroscopic and photophysical properties [26,33,34].

Nevertheless, shifting the absorption maxima to the red region, in the particular case of the coumarin derivatives, is still a significant challenge. The diethylamino electron-donating group (EDG) at position 7 creates an evident push−pull effect with the electron-withdrawing lactone moiety, which induces larger red shifts in the absorption spectrum than other EDG substituents, such as alkoxy groups [35]. In this work, we develop a simple, low-cost and effective synthetic strategy to produce new red-shifted 4-styrylcoumarin derivatives using 7-diethylamino-4-methylcoumarin (**1**) as a starting material, to produce fluorescent labels for biomolecules. Additionally, density functional theory (DFT) and time-dependent density functional theory (TDDFT) calculations are conducted in some of the coumarin derivatives, in order to gain deeper insight into their electronic and spectroscopic properties.

## 2. Results

### 2.1. Synthesis

The main synthetic strategy to obtain 4-styrylcoumarin derivatives was based on the high acidity of the methyl protons present at position 4 in 2-(7-(diethylamino)-4-methyl-2*H*-chromen-2-ylidene)malononitrile (**3**), which enable aldol condensation reactions. The mentioned dicyanomethylene coumarinylmethyl derivative, presenting a higher bathochromic shift (more than 100 nm) when compared with its precursor (**2**), was obtained by the incorporation of two cyano groups in position 2 after the thionation of (**1**) [35,36]. We reasoned that the incorporation of one 4-styryl group containing electron-donating groups (EDGs) or electron-withdrawing groups (EWGs) at the *para* position could increase the π delocalization and the push−pull character of the chromophore. These modifications can promote large bathochromic shifts in the absorption and emission bands, as well as improve other photophysical properties. According to the above, we describe the design, synthesis, and spectroscopic characterization of a small library of 4-styrylcoumarin derivatives to explore the effect of EWGs and EDGs on the photophysical properties of the new chromophores. Moreover, one of the best candidates considering the binomial cost/photophysical properties has been functionalized through a reactive succinimidyl ester as an effective fluorescent label for biomolecules. The synthetic routes followed for the preparation of the novel 4-styrylcoumarin derivatives are shown in Figure 1. The stereoselective and highly efficient aldol condensation reaction between the intermediate (**3**) and the aromatic aldehydes (methyl 4-formylbenzoate, benzaldehyde, 4-methoxybenzaldehyde, and 4-(dimethylamino)benzaldehyde), afforded the 4-styrylcoumarin derivatives (**4** to **7**) in very-good-to-high yields.

Unexpectedly, despite needing stronger reaction conditions, aldehydes containing electron-donating groups gave better yields than the one with electron-withdrawing group at the *para* position, which may be due to the superior stabilization of the intermediate alcohol, by the first ones, in this kind of molecule. The aforementioned new π-extended coumarins (Figure 1) are easily isolated after silica column chromatography. All spectral data were consistent with the proposed structures (Appendix A).

Considering the good photophysical properties induced by the methoxy group in the 4-styrylcoumarin **4**, we selected this compound to synthesize the amine-reactive fluorescent label **9** (Figure 2). The referred fluorescent label can be obtained through five effective and linear synthetic steps from cheap commercially available precursors and, as it is later in the paper, its photophysical properties are very similar to those of derivative **7**. The aldol condensation of **3** with the 6-(4-formylphenoxy)hexanoic acid afforded compound **8**, which was further reacted with *N*-hydroxysuccinimide to attain the fluorescent label **9** (Figure 2).

### 2.2. Photophysical Properties

The photophysical properties of the synthesized coumarin derivatives were studied, and their absorption and emission properties, as well as fluorescence quantum yields, are summarized in Table 1.

The absorption and emission spectra of the new 4-styrylcoumarin derivatives (**4** to **7**) are displayed in Figure 2. All previous mentioned coumarin derivatives exhibit absorption and emission maxima at longer wavelengths, when compared to the intermediate, dicyanomethylene coumarin methyl derivative **3**, due to the intramolecular charge transfer (ICT) effect by the conjugation of both the electron-donating NEt_2_ group and the styryl groups in position 4 with the electron-withdrawing dicyanomethylene group in position 2.

Generally, the presence of electron-withdrawing groups at position 4 promotes higher bathochromic shifts than the electron-donating groups [37], but this effect is not significantly pronounced in the 4-styrylcoumarin derivatives (**4** to **7**), possibly due to the strong electron-withdrawing effect of the dicyanomethylene group in position 2. On the other hand, the molar extinction coefficients were strongly affected by the nature of EDGs or EWGs in the *para* position at the 4-styryl group. The analysis of Table 1 allows the verification of the fact that, in the case of the 4-styrylcoumarin derivatives (**4** to **7**), EDGs promote high coefficients (e.g., ε (**6**) = 34,000 cm^−1^M^−1^ vs. ε (**4**) = 19,000 cm^−1^M^−1^) and also high fluorescence quantum yields (e.g., Φ_F_ (**6**) = 0.95 vs. Φ_F_ (**4**) = 0.04).

All coumarin derivatives exhibit large Stokes shifts due the extension of π-conjugated system in the molecule, which is essential to the effective intramolecular charge transfer process of emissive excited state. The coumarin derivative **6** presents the higher fluorescence quantum yield, possibly due the presence of a strong electron donating amino substituent, but the decrease observed in the derivative **7** might be attributed to a substantial reduction in the oxygen atom electronic density in the presence of the dicyanomethylene electron-withdrawing substituent [38].

Fluorescent labels with large Stokes shifts values offer an advantage due the possible elimination of spectral overlap between absorption and emission, which reduces interference and also eliminates the quenching process, providing a very simple detection of the fluorescence emission. Fluorescent labels with large Stokes shifts are very important for Förster-type resonance energy-transfer (FRET) applications [39] and optical microscopy based on stimulated emission depletion (STED) [40].

### 2.3. Theoretical Calculations

The optimized molecular geometry of the most relevant coumarin derivatives in both the ground and the first excited singlet state computed at PBE0/6-31G(d, p) level in acetonitrile (see the SI for computational details) are depicted in Appendix A and detailed in Appendix A. In the electronic ground state, all compounds present a deviation from planarity between the mean plane of the coumarin rings moiety and the 4-styryl group plane, restraining the π conjugation that links the donor and acceptor groups in the molecules. The ethylene bridge stays nearly in the plane of the benzyl group and the torsion occurs on the bond with the coumarin moiety. This dihedral angle is smaller for compound **6** (19.8°), 19.8° and 19.7° for **7** and **9**, respectively, and presents higher values for compounds **4** (24.5°) and **5** (23.0°). For the excited state S1, however, a completely different picture emerges, with all the molecules becoming nearly planar with a minimum of −0.9° for compounds **4** and maximum of 0.7° for compound **9**. Another important geometrical property that is pertinent to access the electronic delocalization throughout the π-conjugation framework is the bond length alternation (BLA), the average length difference between a single and adjacent double bond. While the BLA of the compounds varies between 0.08 Å (compound **6**) and 0.11 Å (compounds **4** and **5**) in the ground S0 state, it considerably reduces to 0.04 Å for all compound in the S1 excited state. These significant differences in geometry between the relaxed S0 and S1 states, contributing for an enhancement of electronic delocalization and a decrease in the HOMO-LUMO energy gap, might justify a large Stokes shift, where, after a vertical excitation, a significant structural relaxation of the excited state follows prior to emission [41,42].

The absorption wavelengths, oscillator strength (*f*) and the main components of lowest energy transitions of the compounds were calculated by time-dependent density functional theory (TDDFT) methods (Figure 3, Appendix A). The lowest-energy excitations S1 and S2, despite presenting different oscillator strengths, share the same composition for all the compounds and are mainly of HOMO→LUMO and HOMO-1→LUMO character, respectively (more than 98% in all cases). 

The shape and spatial location of the states, however, is very different, which imply different characteristics of the intramolecular charge transfer that occurs upon excitation. In most cases, the HOMO and HOMO-1 are more localized either in the coumarin or the 4-styryl group, while the LUMO are less localized, extending over the bridging zone, which facilitates low energy internal charge transfer absorptions [43]. The lowest energy excitation for compound **6**, which exhibit the transitions with the higher oscillator strength, correspond to charge transfers from the 4-styryl group to the coumarin moiety, reflecting the strong donor capability of the amine group attached to the 4-styryl group. For the other compounds, the S0→S1 excitation corresponds to the opposite charge transfers, with the HOMO state mainly located on the coumarin moiety and the LUMO spreading to the 4-styryl group. For these derivatives, however, this is not the most intense excitation, with the S0→S2 transition presenting much higher f values and thus dominating the absorption spectra. The donor character of compounds **7** and **9** attached units is evidenced by this transition that takes place between the HOMO-1 state, mostly located on the 4-styryl group, and the LUMO, mostly located on the coumarin moiety nitrile groups.

Comparing the MOs of **7** and **9**, they are comparable since the attached reactive group does not take part on the π-conjugation framework, thus resulting in very close spectroscopic properties. For most of the compounds, the calculated lowest energy transitions present good agreement with the experimental absorption maxima of the coumarin derivatives absorption (Table 1 and Appendix A) with the exception of compound **6**. For this compound, the experimental maximum at 496 nm can be assigned to the S0→S2 transition, while the lowest energy S0→S1 transition corresponds to a visible shoulder located around 536 nm.

## 3. Materials and Methods

### 3.1. General Methods

All starting materials and reagents were analytical grade, purchased from Aldrich and used without further purification. The organic solvents were dried over appropriate drying agents and distilled prior to use. UV-Vis absorption spectra were recorded on the Thermo Electron Spectrophotometer Corporation, model Nicolet Evolution 300, using acetonitrile (CH_3_CN) as solvent. Fluorescence measurements of aerated solutions were performed on a PerkinElmer Model LS 55 spectrophotometer. All emission spectra were collected with 5.0 nm slit bandwidth for excitation and emission, with correction files. All spectroscopic measurements were performed in 3 mL quartz fluorescence cuvettes (1 cm optical path) at 21 °C. FTMS-ESI mass spectra were obtained on an Thermo Scientific Q Exactive Orbitrap Mass Spectrometer. Nuclear magnetic resonance (NMR) spectra were recorded at 400 MHz for ^1^H NMR and at 100 MHz for ^13^C NMR, on a Brucker Advance III spectrometer. For NMR spectra, deuterated chloroform (CDCl_3_) was used as solvent. The chemical shift (δ) in ppm; coupling constants (J); relative intensity is indicated by the number of protons (H); multiplicities are indicated by singlet (s), doublet (d), double-doublet (dd), triplet (t), quadruple (q) and multiplet (m); coupling constants are given in Hertz (Hz). Quantum chemical calculations were performed with the Gaussian 16 software pack-age [44]. The hybrid PBE0 functional [45] with the standard 6-31G(d,p) basis set was used for geometry optimizations, both in the ground state and in the first singlet ex-cited state, while the larger 6-311+G(d,p) basis set was employed for the spectra calcu-lations. The PBE0 functional, besides being widely used for calculating ground state geometries, proved also to be very effective for TD-DFT calculation of excited-state structures [46]. In all cases solvent effects were taken into account by means of the im-plicit polarized continum model (PCM) [47,48]. No symmetry constraints are used during geometry optimizations. The frequency analysis confirmed the optimized structures as minima presenting all real valued frequencies.

### 3.2. Procedures for the Preparation of Compounds

#### 3.2.1. 2-(7-(Diethylamino)-4-methyl-2*H*-chromen-2-ylidene)malononitrile (**3**)

The synthesis of dicyanomethylenecoumarinmethyl derivative (**3**) was performed according to the method described by Gandioso et al. [35]

#### 3.2.2. (*E*)-Methyl 4-(2-(2-(dicyanomethylene)-7-(diethylamino)-2*H*-chromen-4-yl)benzoate (**4**)

A mixture of compound **3** (45 mg, 0.161 mmol, 1.0 eq), methyl 4-formylbenzoate (26 mg, 0.161 mmol, 1.0 eq) and piperidine (16 µL, 0.161 mmol, 1.0 eq) in dry acetonitrile (4 mL) was stirred at 40 °C, for a period of 2 h. The reaction was continuously monitored by TLC, using CH_2_Cl_2_ as eluent. After cooling to room temperature, the reaction mixture was evaporated to dryness and the residue was purified by flash chromatography, using CH_2_Cl_2_ as eluent, to yield methyl (*E*)-4-(2-(2-(dicyanomethylene)-7-(diethylamino)-2*H*-chromen-4-yl)benzoate as a dark red solid (48 mg, 71%). mp 252–254 °C. ^1^H NMR (400 MHz, CDCl_3_): 1.25 (6H, t, J = 7.1, N(CH_2_*CH_3_*)_2_), 3.47 (4H, q, J = 7.1, N(CH_2_CH_3_)_2_), 3.94 (3H, s, H-25), 6.59 (1H, d, J_8,6_ = 2.4, H-8), 6.71 (1H, dd, J_6,5_ = 9.2, J_6,8_ = 2.4, H-6), 6.83 (1H, s, H-3), 7.40 (2H, sl, H-9, H-17), 7.63 (1H, d, J_5,6_ = 9.2, H-5), 7.64 (2H, d, J = 8.2, H-19, H-23), 8.08 (2H, d, J = 8.2, H-20, H-22). ^13^C NMR (100 MHz, CDCl_3_): 12.6 (C-11, C-13), 45.1 (C-10, C-12), 52.4 (C-25), 97.7 (C-8), 103.7 (C-3), 108.5 (C-4a), 110.7 (C-6), 114.6 (C-15, C-16), 115.6 (C-14), 122.2 (C-9), 125.8 (C-5), 127.7 (C-19, C-23), 130.4, (C-20, C-22), 131.1 (C-21), 137.4 (C-17), 139.6 (C-18), 146.0 (C-4), 151.8 (C-7), 155.5 (C-8a), 166.6 (C-24) 171.6 (C-2). FTMS(+) calc. for C_26_H_24_N_3_O_3_ [M+H]^+^ 426.1812 found 426.1807. UV λ^max^ (nm, CH_3_CN): 312, 348, 395, 520. ε (cm^−1^ M^−1^): 19,000. Φ_F_ = 0.04.

#### 3.2.3. (*E*)-2-(7-(Diethylamino)-4-styryl)-2*H*-chromen-2-ylidene)malononitrile (**5**)

A mixture of compound **3** (100 mg, 0.358 mmol, 1.0 eq), benzaldehyde (36 µL, 0.358 mmol, 1.0 eq) and piperidine (36 µL, 0.358 mmol, 1.0 eq) in dry acetonitrile (4 mL) was stirred at 85 °C, for a period of 5 min. The reaction was continuously monitored by TLC, using CH_2_Cl_2_/hexane (8:2) as eluent. After cooling at room temperature, the reaction mixture was evaporated to dryness and the residue purified by flash chromatography, using CH_2_Cl_2_/hexane (8:2) as eluent, to yield (*E*)-2-(7-(diethylamino)-4-styryl)-2*H*-chromen-2-ylidene) malononitrile, as a red solid (121 mg, 92%). mp 162–164 °C. ^1^H NMR (400 MHz, CDCl_3_): 1.25 (6H, t, J = 7.1, N(CH_2_*CH_3_*)_2_), 3.47 (4H, q, J = 7.1, N(*CH_2_*CH_3_)_2_), 6.61 (1H, d, J_8,6_ = 2.4, H-8), 6.70 (1H, dd, J_6,5_ = 9.1, J6,8 = 2.4, H-6), 6.87 (1H, s, H-3), 7.34 (1H, d, J_9,17_ = 16, H-9), 7.40–7.46 (4H, m, H-17, H-20, H-22, H-21), 7.60 (2H, d, J_19,20_ = 6.8, H-19, H-23), 7.65 (1H, d, J_5,6_ = 9.1, H-5). ^13^C NMR (100 MHz, CDCl_3_): 12.5 (C-11, C-13), 45.0 (C-10, C-12), 97.6 (C-8), 103.1 (C-3), 108.6 (C-4a), 110.6 (C-6), 114.8 (C-15, C-16), 115.7 (C-14), 119.5 (C-9), 125.7 (C-5), 127.8 (C-19, C-23), 129.1 (C-20, C-22), 130.1 (C-21), 135.4 (C-18), 138.9 (C-17), 146.5 (C-4), 151.6 (C-7), 155.4 (C-8a), 171.6 (C-2). FTMS(+) calc. for C_24_H_22_N_3O_ [M+H]^+^ 368.1757 found 368.1752. UV λ^max^ (nm, CH_3_CN): 341, 396, 516. ε (cm^−1^ M^−1^): 17,000. Φ_F_ = 0.16.

#### 3.2.4. (*E*)-2-(7-(Diethylamino)-4-(4-(dimethylamine)Styryl)-2*H*-chromen-2-ylidene)malononitrile (**6**)

A mixture of compound **3** (150 mg, 0.537 mmol, 1.0 eq), 4-(dimethylamine)benzaldehyde (81 mg, 0.537 mmol, 1.0 eq) and piperidine (60 µL, 0.536 mmol, 1.0 eq), in dry acetonitrile (4 mL), was stirred at 85 °C, for a period of 24h. The reaction was continuously monitored by TLC, using CH_2_Cl_2_ as eluent. After cooling to room temperature, the reaction mixture was evaporated to dryness and the residue was purified by flash chromatography, using as eluents CH_2_Cl_2_/hexane (8:2), CH_2_Cl_2_/hexane (9:1), and CH_2_Cl_2_ to yield (*E*)-2-(7-(diethylamino)-4-(4-(dimethylamine)styryl)-*2H*-chromen-2-ylidene)malononitrile, as a red solid (210 mg, 95%). mp 214–216 °C. ^1^H NMR (400 MHz, CDCl_3_): 1.24 (6H, t, J = 7.1, N(CH_2_*CH_3_*)_2_), 3.06 (6H, s, 2xNCH_3_), 3.45 (4H, q, J = 7.1, N(*CH_2_*CH_3_)_2_), 6.54 (1H, d, J_8,6_ = 2.4, H-8), 6.68 (2H, d, J_20,19_ = 8.8, H-20, H-22), 6.69 (1H, dd, J_6,5_ = 8.8, J_6,8_ = 2.4, H-6), 6.77 (1H, s, H-3), 7.08 (1H, d, J_9,17_ = 15.8, H-9), 7.37 (1H, d, J_17,9_ = 15.8, H-17), 7.47 (1H, d, J_19,20_ = 8.8, H-19, H-23), 7.66 (1H, d, J_5,6_ = 8.8, H-5). ^13^C NMR (100 MHz, CDCl_3_): 12.7 (C-11, C-13), 40.3 (C-24, C-25), 45.0 (C-10, C-12), 97.6 (C-8), 100.8 (C-3), 108.8 (C-4a), 110.5 (C-6), 112.1 (C-20, C-22), 113.2 (C-9), 115.7 (C-14), 116.6 (C-15, C-16), 123.3 (C-18) 125.6 (C-5), 129.9 (C-19, C-23), 140.0 (C-17), 147.3 (C-4), 151.5 (C-7), 151.8 (C-21), 155.5 (C-8a), 171.3 (C-2). FTMS(+) calc. for C_26_H_27_N_4_O [M+H]^+^ 411,2179 found 411,2174. UV λ^max^ (nm, CH_3_CN): 294, 341, 496. ε (cm^−1^ M^−1^): 34,000. Φ_F_ = 0.95.

#### 3.2.5. (*E*)-2-(7-(Diethylamino)-4-(4-methoxystyryl)-2*H*-chromen-2-ylidene)malononitrile (**7**)

A mixture of compound **3** (279 mg, 0.251 mmol, 1.0 eq), 4-methoxybenzaldehyde (30 µL, 0.251 mmol, 1.0 eq) and piperidine (25 µL, 0.251 mmol, 1.0 eq) in dry acetonitrile (5 mL) was stirred at 85°C, for a period of 17h. The reaction was continuously monitored by TLC, using CHCl_3_/CH_3_OH (99:1) as eluent. After cooling to room temperature, the reaction mixture was evaporated to dryness and the residue was purified by flash chromatography, using as eluent CHCl_3_/CH_3_OH (99:1) to yield (*E*)-2-(7-(diethylamino)-4-(4-metoxystyryl)-2*H*-chromen-2-ylidene)malononitrile, as a red solid (95 mg, 96%). mp 230–232 °C. ^1^H NMR (400 MHz, CDCl_3_): 1.24 (6H, t, J = 7.1, N(CH_2_*CH_3_*)_2_), 3.46 (4H, q, J = 7.1, N(*CH_2_*CH_3_)_2_), 3.87 (3H, s, H-24), 6.58 (1H, d, J_8,6_ = 2.4, H-8), 6.70 (1H, dd, J_6,5_ = 9.1, J_6,8_ = 2.4, H-6), 6.82 (1H, s, H-3), 6.94 (2H, d, J_20,19_ = 8.7, H-20, H-22), 7.19 (1H, d, J_9,17_ = 15.9, H-9), 7.39 (1H, d, J_17,9_ = 15.9, H-17), 7.54 (1H, d, J_19,20_ = 8.7, H-19, H-23), 7.65 (1H, d, J_5,6_ = 9.1, H-5). ^13^C NMR (100 MHz, CDCl_3_): 12.6 (C-11, C-13), 45.1 (C-10, C-12), 55.6 (C-24), 97.7 (C-8), 102.4 (C-3), 108.7 (C-4a), 110.6 (C-6), 114.7 (C-20, C-22), 115.1 (C-14), 116.1 (C-15, C-16), 116.9 (C-9), 125.8 (C-5), 128.3 (C-18), 129.6 (C-19, C-23), 138.8 (C-17), 146.9 (C-4), 151.7 (C-7), 155.6 (C-8a), 161.5 (C-21), 171.6 (C-2). FTMS(+) calc. for C_25_H_24_O_2_N_3_ [M+H]^+^ 398,1863 found 398,1854. UV λ^max^ (nm, CH_3_CN): 418, 522. ε (cm^−1^ M^−1^): 24,000. Φ_F_ = 0.20.

#### 3.2.6. (*E*)-6-(4-(2-(2-(Dicyanomethylene)-7-(diethylamino)-2*H*-chromen-4-yl)vinyl)phenoxy)hexanoic Acid (**8**)

A mixture of compound **3** (125 mg, 0.448 mmol, 1.0 eq), 6-(4-formylphenoxy)hexanoic acid (106 mg, 0.448 mmol, 1.0 eq) and piperidine (44 µL, 0.448 mmol, 1.0 eq) in dry acetonitrile (10 mL) was stirred at 85 °C for 24 h. The reaction was monitored by TLC using CHCl_3_/CH_3_OH (95:5) as eluent. After cooling to room temperature, the reaction mixture was evaporated to dryness and the residue was purified by flash chromatography, using CHCl_3_/CH_3_OH (99:1) and CHCl_3_/CH_3_OH (95:5) as eluents, to yield (*E*)-6-(4-(2-(2-(dicyanomethylene)-7-(diethylamino)-2*H*-chromen-4-yl)vinyl)phenoxy)hexanoic acid, as a red solid (187 mg, 84%). ^1^H NMR (400 MHz, CDCl_3_): 1.25 (6H, t, *J* = 6.8, N(CH_2_*CH_3_*)_2_), 1.56 (2H, m, H-26), 1.74 (2H, m, H-27), 1.84 (2H, m, H-25), 2.42 (2H, t, *J* = 6.6, H-28) 3.46 (4H, q, *J* = 6.8, N(*CH_2_*CH_3_)_2_), 4.02 (2H, t, H-24), 6.58 (1H, sl, H-8), 6.70 (1H, d, *J*_6,5_ = 9.2, H-6), 6.83 (1H, s, H-3), 6.93 (2H, d, *J*_20,19_ = 7.8, H-20, H-22), 7.19 (1H, d, *J*_9,17_ = 15.8, H-9), 7.39 (1H, d, *J*_17,9_ = 15.8, H-17), 7.53 (2H, d, *J*_19,20_ = 7.8, H-19, H-23), 7.65 (1H, d, J_5,6_ = 9.2, H-5).^13^C NMR (100 MHz, CDCl_3_): 12.7 (C-11, C-13), 24.5 (C-27), 25.7 (C-26), 29.0 (C-25), 33.8 (C-28), 45.1 (C-10, C-12), 68.0 (C-24), 97.7 (C-8), 102.3 (C-3), 108.7 (C-4a), 110,6 (C-6), 115.1 (C-14), 115.2 (C-20, C-22), 116.1 (2xCN), 116.8 (C-9), 125.8 (C-5), 128.2 (C-18), 129.6 (C-19, C-23), 138.9 (C-17), 147.0 (C-4), 151.7 (C-7), 155.6 (C-8a), 160.9 (C-21), 171.6 (C-29), 178.7 (C-2). FTMS(+) calc. for C_30_H_32_N_3_O_4_ [M + H]^+^ 498.2387 found 498.2379. UV λ^max^ (nm, CH_3_CN): 340, 418, 519. ε (cm^−1^ M^−1^): 24,000. Φ_F_ = 0.28.

#### 3.2.7. (*E*)-2,5-Dioxopyrrolidin-1-yl 6-(4-(2-(2-(dicyanomethylene)-7-(diethylamino)-2*H*-chromen-4-Yl)vinyl)phenoxy)hexanoate (**9**)

A mixture of compound 8 (110 mg, 0.221 mmol, 1.0 eq), DCC (55 mg, 0.265 mmol, 1.2 eq), DMAP (3 mg, 0.0221 mol, 10%) in dry DMF (2 mL) and dry acetonitrile (25 mL) was stirred at room temperature for 5 min. After this period, *N*-hydroxysuccinimide (28 mg, 0.243 mmol, 1.2 eq) was added and the reaction was stirred at room temperature for 5 h. The reaction was monitored by TLC using CHCl_3_/CH_3_OH (95:5) as eluent. The reaction mixture was evaporated to dryness and the residue was purified by flash chromatography using CH_3_CN/CHCl_3_ (7:3) as eluents, to yield (E)-2,5-dioxopyrrolidin-1-yl 6-(4-(2-(2-(dicianomethylene)-7-(diethylamino)-2*H*-chromen-4-yl)vinyl)phenoxy)hexanoate, as a red solid (117 mg, 89%). mp 192–194 °C. ^1^H NMR (400 MHz, CDCl_3_): 1.24 (6H, t, *J* = 6.8, N(CH_2_*CH_3_*)_2_), 1.58–1.66 (2H, m, H-26), 1.83–1.88 (4H, m, H-25, H-27), 2.67 (2H, t, *J* = 7.4, H-28), 2.85 (2H, m, H-31, H-32), 3.46 (4H, q, *J* = 6.8, N(*CH_2_*CH_3_)_2_), 4.02 (2H, t, *J* = 6.4, H-24), 6.59 (1H, d, *J*_8,6_ = 2.4, H-8), 6.70 (1H, dd, *J*_6,5_ = 9.2, *J*_6,8_ = 2.4, H-6), 6.85 (1H, s, H-3), 6.93 (2H, d, *J*_20,19_ = 8.6, H-20, H-22), 7.20 (1H, d, *J*_9,17_ = 16.0, H-9), 7.40 (1H, d, *J*_17,9_ = 16.0, H-17), 7.54 (2H, d, *J*_19,20_ = 8.6, H-19, H-23), 7.66 (1H, d, *J*_6,5_ = 9.2, H-5). ^13^C NMR (100 MHz, CDCl_3_): 12.6 (C-11, C-13), 24.5 (C-27), 25.4 (C-26), 25.7 (C-31, C-32), 28.8 (C-25), 31.0 (C-28), 45.1 (C-10, C-12), 67.8 (C-24), 97.7 (C-8), 102.4 (C-3), 108.8 (C-4a), 110,6 (C-6), 115.2 (C-14, C-20, C-22), 116.8 (2xCN, C-9), 125.8 (C-5), 128.2 (C-18), 129.6 (C-19, C-23), 138.9 (C-17), 147.0 (C-4), 151.7 (C-7), 155.6 (C-8a), 160.9 (C-21), 168.6 (C-29), 169.3 (C-30, C-33), 171.7 (C-2). FTMS(+) calc. for C_34_H_35_N_4_O_6_ [M + H]^+^ 595.2551 found 595.2543. UV λ^max^ (nm, CH_3_CN): 340, 418, 523. ε (cm^−1^ M^−1^): 24,000. Φ_F_ = 0.29.

## 4. Conclusions

With the objective to extend the delocalization of the π-electron system, we designed and synthesized new 4-styrylcoumarin derivatives, with absorption and emission at long wavelengths, combined with large Stokes shifts, using a simple, low-cost and efficient synthetic strategy. The results obtained from the UV/Vis spectra allow to conclude that the 2-(7-(diethylamino)-4-methyl-2*H*-chromen-2-ylidene)malononitrile (**3**) is a useful intermediate to the synthesis of promising new fluorescent labels for biomolecules. DFT and TDDFT calculations helped to rationalize the observed photophysical properties, particularly the large Stokes shifts, ascribed to a significant structure relaxation found in the molecules’ excited states. The development of further red-shifted coumarin fluorescent labels for biomolecules with improved features is currently in progress in our laboratory and the results will be reported briefly.

## Data Availability

The data presented in this study are available in this article and in the Supporting Information.

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
