# Peer review of "New Red-Shifted 4-Styrylcoumarin Derivatives as Potential Fluorescent Labels for Biomolecules"

_molecules, 2022, doi:10.3390/molecules27051461_

Round 1

Reviewer 1 Report

Overall comment: The present manuscript reports the synthesis, photophysical and theoretical calculations studies on a series of 4-styrylcoumarin derivatives as potential fluorescent labels for biomolecules. All the synthesized compounds were fully characterized by 1H and 13C NMR spectra and HRMS. The structural diversity of the synthesized coumarins allowed the authors to understand the influence on the absorption-emission as well as fluorescence quantum yields of the diverse substituents.

The manuscript is suitable for publication after minor corrections such as:

Line 59 There is the mention of a compound 1, the 7-diethylamino-4-methylcoumarin, but it structure should be present in Scheme 1

Line 60 Move the sentence “The diethylamino electron-donating group (EDG) at position 7 creates an evident push−pull effect with the electron-withdrawing lactone moiety, which induces larger red shifts in the absorption spectrum than other EDG substituents such as alkoxy groups [35]” to line 57 before the sentence that starts with the following “In this work, we….”

Line 71 – Try to rephrase the following sentence “The mentioned dicyanomethylene-coumarinylmethyl derivative, with a higher bathochromic shift than 100 nm when compared with their precursor, was obtained by the incorporation of two cyano groups in position 2 after the thionation of the carbonyl group of the lactone.” to the following since the English is a little bit incomprehensive. “The mentioned dicyanomethylene-coumarinylmethyl derivative, presenting a higher bathochromic shift (more than 100 nm) when compared with its precursor (2), was obtained by the incorporation of two cyano groups in position 2 after the thionation of (1)”

Line 76, 97,134 “para” to para (in italic)

Line 85 remove (4 to 7) since its repeating in line 91

Line 86-88 Remove the sentence “The carbonyl group thionation….in high yields [35]” Its already mentioned previously.

Line 97 and 98 what is the intermediate aldol mentioned here by the authors?

Line 101 Figure 1. The legend of the figure should mention the numbers in the figure “Photographic images of coumarin derivatives 1, 3-9) in MeCN at 365 nm.

And is 365 or 336 nm?

Line 104 “….4-styrylcoumarin derivatives we selected this substituent to synthesize an inexpensiveand effective amine-reactive fluorescent label for biomolecules” should be “….4-styrylcoumarin 4 we selected this compound to synthesize the amine-reactive fluorescent label 9.”

Line 106 “…fluorescent label can be obtained..” change to “…fluorescent label 9 can be obtained..”

Line 107 “….cheap commercially available precursors and as already mentioned, its photophysical properties are very similar to the derivative 7” change to “….cheap commercially available precursors and as it will be shown ahead, its photophysical properties are very similar to the derivative 7

Also a brief paragraph should be added (before scheme 2) describing the synthetic steps as the reagents used to attain compound 9. Something as “The aldol condensation of 3 with the 6-(4-formylphenoxy)hexanoic acid afforded compound 8 which was further reacted with N-hydroxysuccinimide to attain the fluorescent label 9 (scheme 2)”

Line 123 “…due to the intramolecular charge transfer (ICT) effect between the electron-donating NEt2 group conjugated with the styryl groups in position 4 and the electron-withdrawing dicyanomethylene group in position 2” change to “…due to the intramolecular charge transfer (ICT) effect by conjugation of both the electron-donating NEt2 group and the styryl groups in position 4 with the electron-withdrawing dicyanomethylene group in position 2

Line 130 “…observed in the 4-styrylcoumarin derivatives (4 to 7)” change to “…so pronounced in the 4-styrylcoumarin derivatives (4 to 7)”

Line 230, 246, 263, 282, 299, 321, what is ccd? Something recurrently mentioned in the experimental procedure.

Line 325, 326 Check the NMR attribution of compound 9. Is not in accordande with the spectra.

Reviewer 2 Report

This manuscript describes ongoing studies by Pereira and coworkers on the design, synthesis, and spectroscopic characterization of a series of push-pull chromophores based on coumarin scaffolds. The synthetic strategy is based on a Knoevenagel condensation of a coumarinylmethyl derivative with the corresponding aryl-aldehydes containing electron-withdrawing or electron-donating groups. In addition, the route has been proven to be amenable to structural modifications. Thus, allowing the attachment of molecular handles. For instance, a reactive succinimidyl ester was prepared to show the potential of these derivatives as fluorescent labels of nitrogen-containing biomolecules. The strategy developed in this manuscript is simple and might prove to be useful. Consequently, in my opinion, the article is worthy of publication.

There are however some issues that need to be addressed before publication:

In the Introduction section: please consider replacing “Coumarins….constitute the main class of fluorescent dyes used in research focused…..” to “Coumarins….are interesting alternatives for applications in …” since the former assumption might be overstated.

The introduction of a styryl-group in a 7-diethylamino-4-methylcoumarin derivative through a Knoevenagel reaction with aldehydes has some literature precedent: Sensors & Actuators: B. Chemical 320 (2020) 128360. This work should be acknowledged in the reference section.

On the other hand, coumarins are usually crystalline compounds. The authors have not included in the manuscript whether the new compounds are solid, crystalline or amorphous, lacquer type solids, etc  Please consider including in the manuscript these characteristics. In the case the compounds appear as solids, please include some melting points.

Reviewer 3 Report

In this manuscript, the authors report the design, synthesis, photophysical properties, and electronic structures of a series of coumarin derivatives that exhibit red fluorescence. The authors also synthesized a bioconjugatable coumarin dye that enables covalently attachment to biomolecules.

Comments:

  • The authors are encouraged to include the considerations of brightness of the coumarin dyes compared to the existing fluorophores such as rhodamines or Alexa Fluor Dyes in the results or discussions section since brightness is a key parameter in fluorescence-based bioimaging and other applications. The fluorescence quantum yield is crucial but not the only factor that contributes to the performance of the fluorescent dye applications in life sciences.
  • The authors need to elaborate more on why they choose the coumarin 7 over 6 as an ideal target to develop a bioconjugation handle. From the photophysical properties of these two dyes, the dye 6 has higher fluorescence quantum yield as well as higher oscillator strength compared to 7. I don’t see why 7 is superior to 6 in terms of the data shown in the manuscript.
  • The authors discussed that “In most cases, the decrease of fluorescence quantum yield in the molecules is directly related with the effective intramolecular charge transfer process due to the extension of conjugated system.” I am not sure if the authors are referring to the excited-state properties of coumarin or common fluorophores. The diminished radiative decay as emission may stem from several factors such as the polarity of solvents and other nonradiative decays such as ISC and IC other than the decay pathway to a charge-transfer state. The authors are encouraged to discuss more on this or to clarify this argument.
  • One of the requirements for labelling fluorescent dyes to biomolecules is the water-solubility of the fluorophore. Many organic fluorophores exhibit great brightness in organic solvents but suffer diminished fluorescence in water due to various quenching mechanisms. The authors are encouraged to include the experimental data how the current designs of the coumarin dyes behave in aqueous media (such as in micellar solution) or high polar solvents to evaluate the photophysical properties in biological environments.
  • Figure 3 is hard to read due to low resolution.

In summary, this manuscript should be reconsidered after a major revision before it can be recommended for publication in the Molecules.

Round 2

Reviewer 3 Report

The manuscript after revision is recommended for publication in Molecules.